# Antitumor Effect of Cycloastragenol in Colon Cancer Cells via p53 Activation

**DOI:** 10.3390/ijms232315213

**Published:** 2022-12-02

**Authors:** Doil Park, Ji Hoon Jung, Hyun Min Ko, Wona Jee, Hyungsuk Kim, Hyeung-Jin Jang

**Affiliations:** 1College of Korean Medicine, Kyung Hee University, Seoul 02447, Republic of Korea; 2Department of Korean Medicine Rehabilitation, Kyung Hee University Medical Center, Seoul 02447, Republic of Korea

**Keywords:** p53, apoptosis, cycloastragenol, colon cancer, 5-FU

## Abstract

Colorectal cancer cell (CRC) is the fourth most common cancer in the world. There are several chemotherapy drugs available for its treatment, though they have side effects. Cycloastragenol (CY) is a compound from Astragalus membranaceus (Fisch.) Bge known to be effective in aging, anti-inflammatory, anticancer, and anti-heart failure treatments. Although many studies have demonstrated the functions of CY in cancer cells, no studies have shown the effects of p53 in colon cancer cells. In this study, we found that CY reduces the viability of colon cancer cells in p53 wild-type cells compared to p53 null cells and HT29. Furthermore, CY induces apoptosis by p53 activation in a dose- and time-dependent manner. And it was confirmed that it affects the L5 gene related to p53. Additionally, CY enhanced p53 expression compared to when either doxorubicin or 5-FU was used alone. Altogether, our findings suggest that CY induces apoptosis via p53 activation and inhibits the proliferation of colon cancer cells. In addition, apoptosis occurs in colon cancer cells due to other factors. Moreover, CY is expected to have a combined effect when used together with existing treatments for colon cancer in the future.

## 1. Introduction

Colorectal cancer is the fourth most common cancer responsible for death in humans. The incidence of colorectal cancer is increasing mainly in developed countries [1]. Although the survival rate of colon cancer is increasing, the treatment is still limited, so it is urgent to find a new treatment. Colorectal cancer develops in a gradual, multistep process involving genetic changes in key tumor suppressors and oncogenes [2].

One of the well-known tumor suppressors, p53, is activated by numerous factors in cancer cells and is deeply involved in cancer cell growth [3,4]. p53 plays an important role in cancer cell growth and apoptosis as a key regulator. The expression of p53 increases in response to various stress signaling, such as hypoxia, DNA damage, ribosomal stress, etc. When p53 is activated, it affects a variety of target genes associated with apoptosis such as p21, Bax, and Puma [5,6]. For example, Bax is associated with improved quality of life of colon cancer patients [7,8]. As already demonstrated in several papers, p53–Bax plays an important role in reducing cancer progression and plays a major role in the treatment and prevention of cancer. Furthermore, PUMA also plays an important role in inducing apoptosis in cancer cells as a target gene of p53 [9].

Astragalus is one of the most widely used traditional Chinese medicinal herbs [10]. Looking at the papers, it is traditionally prepared from the dried root of Astragalus membranaceus Bge. Variables mongholicus (Bge.) Hsiao (A. mongholicus) and Astragalus membranaceus (Fisch.) Bge (A. membranaceus). A. mongholicus and A. membranaceus are legumes, compiled in the Chinese Pharmacopoeia, and mainly exported to Europe. Astragalus has long been mixed with several effective ingredients to make some drugs in the treatment of chronic diseases [11]. Cycloastragenol (CY), the biologically active triterpene aglycone of astragaloside IV, is known at the root of Astragalus membranaceus Bge [12]. Based on published scientific evidence, it has been reported that CY has a variety of medicinal effects associated with anti-inflammatory (inhibition of CD69, CD25, and TXNPI/NLRP3 inflammasomes) [13] and anti-heart failure treatment [14] and anti-aging, [15] in addition to stimulating telomerase activity [16]. Many bioactive compounds derived from other natural resources have already shown tumor potential in several papers. It is also known that cycloastrazenol inhibits compositional STAT3 activation and promotes parklitaxel-induced apoptosis in human gastric cancer cells [17]. However, the mechanism of CY in colon cancer cells by activating p53 is not yet fully understood. Thus, we conducted a study to determine the signaling mechanisms affected by CY in inhibiting the growth of colon cancer cells via activating p53.

## 2. Results

### 2.1. Cycloastragenol Inhibits Colon Cancer Cell Viability

First of all, we investigated whether CY (Figure 1A) suppresses the cell viability of HCT116^p53+/+^, HCT116^p53−/−^, and HT29 cells. MTT assay was performed to determine the cell viability. As shown in Figure 1B, CY inhibits cancer cell viability in a dose-dependent manner. It has been confirmed that HCT116^p53+/+^ cells have decreased cell survival at 50uM, but not at HCT116^p53−/−^ cells. Through this, we confirmed that 50 μM was a suitable concentration for CRC. Interestingly, CY inhibited cancer cell growth in p53 wild-type cells. Consistently, HCT116^p53+/+^ treated with CY dose-dependently interfered with colony formation but did not proliferate at 50 μM in HCT116^p53−/−^ cells (Figure 1C).

### 2.2. Cycloastragenol Induces p53 and Decreases Caspase 3 Expression

The MTT assay results show that CY controls cancer cell growth dependent on p53. To confirm how CY regulates the cancer cell survival, we performed Western blotting and qRT-PCR following treatment with CY for 24 h. As a result, it was observed that p53 increased in a dose-dependent manner in HCT116^p53+/+^ cells (Figure 2A,B). Interestingly, CY decreased the expression of Pro-PARP in HCT116^p53+/+^ cells but not in HCT116^p53−/−^ cells (Figure 2C). Furthermore, to check whether CY induced apoptosis, a TUNEL assay was performed in CY treated HCT116^p53+/+^ cells (Figure 2D).

### 2.3. Cycloastragenol Induces Time-Dependent Apoptosis in HCT116^p53+/+^ Cells

Next, we check whether CY induced apoptosis by time-dependent activation of p53. As shown in Figure 3A, CY induced p53 and its target gene p21 expression. Moreover, CY induced cleaved-PARP expression time-dependently. Consistently, CY increased the expression of Puma and Bax, the target genes of p53 and proteins associated with apoptosis (Figure 3B).

### 2.4. p53 Stability Was Confirmed by Treatment with Cycloheximide in the HCT116^p53+/+^ Cells

Next, we determine how CY regulates p53 protein level by cycloheximide assays. As shown in Figure 4, the CY drug treatment group did not change the half-life of p53 compared to the untreated group. This confirmed that p53 is biochemically stable.

### 2.5. The Role of p53 in CY-Mediated Apoptosis

Next, we tried to confirm whether CY activates p53 to induce apoptosis was conducted using PFT-α, the inhibitor of p53. It was confirmed that CY and PFT-α treatment group attenuated the cleaved-PARP expression as compared to the CY only treated group (Figure 5A). Next, it was checked whether p53 is activated and affects PUMA and p21 in the transcription part using PFT-α. As a result, looking at Figure 5B, it can be confirmed that CY induced an increase in PUMA and p21 in the group treated with PFT-α. Through this, it was confirmed that CY induces apoptosis through PUMA and p21 through p53.

### 2.6. Combinational Effect with CY and 5-FU or Doxorubicin

5-FU and doxorubicin are used as anti-cancer drugs in the treatment of colorectal cancer. However, there is a problem in that various side effects occur when they are used. Therefore, it is urgent find a new drug that can be used in combination with these compounds to act as an anti-cancer drug. To confirm the combination effect of CY with either 5-FU or doxorubicin, we treated CY with or without 5-FU or doxorubicin in HCT116^p53+/+^ cells. The cell viability of the group treated with CY and doxorubicin was observed to be lower than that of the group treated with CY or doxorubicin alone (Figure 6A). Similarly, it was observed that the cell viability was further decreased in the group treated with CY or 5-FU than in the group treated with CY only (Figure 6B). Next, we tested whether CY enhanced p53 expression with doxorubicin and 5-FU in a dose-dependent manner in colon cancer cells (Figure 6C,D).

### 2.7. Ribosomal Protein RPL5 Mediates p53 Activation Due to CY in HCT116^p53+/+^ Cells

In regard to the previous results, we confirmed that CY inhibits apoptosis by increasing p53. Hence, to confirm this once more, the ribosomal protein L5, a gene related to p53, was identified. Looking at Figure 7, it can be confirmed that cells normally increase p53 through the increase of p53 and p21 in the CY-treated group. In addition, it was confirmed that p53 decreased in the siL5-treated group compared to the CY-treated group. Moreover, it was confirmed that CY inhibited apoptosis through the L5-p53 pathway through the decrease in the amount of cleaved PARP.

### 2.8. CY Alleviated Induced 5-FU Resistance of CRC Cells

The development of tumor cell resistance to first-line chemotherapy drugs results in tumor regrowth, metastasis, and poor results. Therefore, drugs that can prevent resistance have high clinical value. Therefore, we processed 5-FU to HCT116^p53+/+^ cells to create 5-FU resistance (Res) cells and proceeded with the experiment. The group trained with HCT116^p53+/+^ cells was created over a long period of time using 5-FU. This process can be referred to other papers mentioned [18]. Looking at Figure 8, it was confirmed that there was no difference in cell viability when comparing the group treated with CY to HCT116^p53+/+^ 5-FU Res cells and the group treated with HCT116^p53+/+^ cells. Through this, it was confirmed that CY is effective in preventing resistance.

### 2.9. CY Induces DNA Damage and Activation of ERK Kinase, p38 MAPK, and JNK

Based on the above study, we found that CY induces apoptosis through p53 activation. However, it was attempted to confirm whether such CY induces apoptosis of p53 through an intercellular stress reaction. Therefore, MAPKs were checked through Western blot. As a result, it was confirmed that p-JNK, p-ERK, and p-p38 were all activated through CY (Figure 9A). Further, we treated SB203580, a p38 inhibitor, to confirm whether p38 affects p53. As a result, it was confirmed that p53 decreased in the group in which p38 was suppressed (Figure 9B). In addition, we used H2AX, a representative DNA damage marker, to confirm whether the activity of these MAPKs induces DNA damage. As a result, an increase in H2AX was observed in the CY-treated group (Figure 9C). Through these results, we were able to confirm that p53 was activated by CY receiving DNA damage through the MAPK pathway.

## 3. Discussion

p53 has been identified as tumor suppressor in cancer cells. Various stress induce activation of p53 such as ribosomal stress, hypoxic condition etc. p53 causes apoptosis, ferroptosis, cell cycle arrest, and inhibit the metastasis in cancer cells. Many genes act as regulators of p53, and there are also various drugs that cause p53 activation, such as 5-FU and doxorubicin.

Cycloastragenol (CY), the biologically active triterpene aglycone of astragaloside IV, is known at the root of Astragalus membranaceus Bge [12]. According to paper, a rapid and highly sensitive UPLC–MS/MS analysis method was used and applied to the simultaneous detection of flavonoids and triterpenoids, Cycloastragenol, Astragaloside I, Astragaloside III and Astragaloside IV in the five different parts of two varieties Astragalus plants (roots, stems, leaves, petioles and flowers) and during 2-,5-year growth periods of A. membranaceus [19]. Three or four of these substances were screened and it was confirmed that CY had the greatest anti-cancer effect.

To our best knowledge, this is the first study to report CY-induced apoptosis through p53 activation in colon cancer cells. Our cell-based studies show that CY inhibits colon cancer cell viability via p53 in a dose-dependent manner. To check whether CY induced apoptosis by p53 activation, we treated CY in HCT116^p53+/+^ or HCT116^p53−/−^ cells. As a result, it was confirmed that CY induced apoptosis depends on p53 activation. Furthermore, we found that CY induced stability of p53 protein levels compared with the control group. In addition, as a result of CY treatment of HCT116^p53+/+^ cells with the p53 inhibitor PFT-α, cleaved-PARP was decreased compared to the CY only treated group, indicating that CY induced apoptosis via activating p53.

As introduced in several papers, us3 represents pro-apoptosis function in addition to its role in DNA repair. The Co-Ip experiment shows that these apoptosis effects are due to physical interactions between us3 and tumor necrosis factor receptor type 1 related death domain protein (TRADD). Further, it is known that pro-apoptosis signals mediated by us3 are executed through the activation of caspase-8/caspase-3 and c-Jun N-terminal kinase (JNK) pathways. DNA repair and apoptosis, two of these functions of us3, include independent functional domains [20]. Several studies have reported S14 (S29) as an inducer of apoptosis in other human cancer cell lines. Enhanced expression of us14 in laryngeal cancer cells induces activation of ap38 MAPK and JNK signals, leading to apoptosis activation. It has been proposed that us14 exerts apoptosis function through death receptor-mediated and mitochondrial-mediated activation of the apoptosis pathway [21]. H2AX phosphorylation patterns are implicated in determining whether cells repair damaged DNA for survival or undergo apoptosis [22]. Results from H2AX knockout studies in mice show that loss of one or both alleles of the H2AX gene impairs genome integrity and DDR efficiency and increases tumorigenicity in a p53-null background [23,24,25,26]. Through this, we confirmed that CY induces DNA damage through MAPKs using H2AX (Figure 9).

Fouracil (5-FU) and doxorubicin are used as treatments for colon cancer patients. 5-FU is a basic component of the chemotherapy for alleviation and auxiliary treatment of CRC. Some 5-FU based treatments, such as FOLFOX (5-FU, leucoborin, oxaliplatin) or FOLFIRI (5-FU, leucoborin, irinotecan), have been used as standard treatments in, e.g., advanced CRC (Sinicrope, F.A). Although the 5-FU-based treatment increased the objective response rate of CRC patients to 40–50%, the disease-free survival rate of CRC patients was not effectively extended [27]. Furthermore, there are various side effects such as nausea, vomiting, and diarrhea. Antracycline drugs such as doxorubicin (DOX) are also most commonly used in the treatment of various cancer cell lines. However, various side effects, such as hair loss, vomiting, rash, canker soreness, and heart damage, occur. In addition to these side effects, problems with resistance arise when used for a long time.

A wide range of stress stimuli (radiation, tumor genes, nutrient deficiency, hypoxia, and genotoxic compounds) can interfere with ribosomal biosynthesis and activate complex cellular reactions, nuclear stress. Nuclear stress pathway activation mediated by multiple nuclei and RP results in cell cycle blocking, activation of apoptosis, DNA damage, and aging. The RNP-MDM2-p53 pathway is activated by stress stimulation, which causes some RP to be released from nucleosomes to nucleoplasma. The interaction between RP and MDM2 causes p53 stabilization and subsequent cell cycle blocking [28,29,30]. Overall, the extra-ribosomal function of RPs regulates affects p53 pathway by regulating several cell processes, including cell cycle, DNA repair, maintaining genome integrity, cell proliferation, apoptosis, cell migration, and invasion [28,31,32].

Furthermore, these data showed that uL3 autoregulates its expression [33,34] and uL3 status is highly identified in cellular responses to specific anticancer agents in p53 mutant lung and p53 deletion colon cancer cells. Notably, uL3 downregulation is positively correlated with multidrug resistance. This protein affects p21 activity independently of p53 in response to chemotherapy-induced nuclear stress [30,35,36]. Moreover, it is known that altered expression of a single RP also helps to alter the ribosomal translation efficiency of certain mRNAs, including transcripts of other RPs or transcripts involved in key regulatory steps of tumorigenesis and drug resistance [37,38,39,40]. Ribosomal protein L5 was used to confirm the inhibition of apoptosis caused by CY-induced increase in p53. RPL5 is a representative gene known to regulate p53 in several paper [41]. We used this to confirm that CY inhibits apoptosis through RPL5-p53 (Figure 7). In the 5-FU resistance cell line data, we confirmed that a similar pattern of drug effect was found in 5-FU Res cells as compared to the control group as a result of treating each cell with CY. Through this, it was confirmed that CY was effective even in 5-FU-resistant cells (Figure 8).

To solve these shortcomings, we treated DOX and 5-FU together with CY to see the effects. Therefore, it is necessary to discover treatments that reduce these side effects above all else. Interestingly, CY enhanced the anticancer effect with 5-FU or doxorubicin in HCT116^p53+/+^ cells. CY induced p53, demonstrating a combination effect when used together with 5-FU or doxorubicin (Figure 6A–D). This result suggests the possibility of parallel administration with existing treatments as a new drug for colorectal cancer cells (Figure 10).

## 4. Materials and Methods

### 4.1. Reagents

We purchased this reagent from ChemFaces (Wuhan ChemFaces Biochemical Co., Ltd., Wuhan, China) for Cycloastragenol (CY). In addition, it was used after concentration and dilution with DMSO. Primary antibodies against Cleaved-caspase3 (cat. 9661s), PARP (cat. 9542s), p-JNK (cat. 9251s), JNK (cat. 9252s), p-ERK (cat. 9101s), ERK (cat. 9102s), p-p38 (cat. 4511s), p38(cat. 9212s), H2AX (2577s) were purchased from Cell Signaling Technology (Beverly, MA, USA). P21 (cat. Sc-817), Bax (cat. Sc-7480), Puma (cat. Sc-374223), p53 (cat. sc-126), β-actin and caspase 3 (sc-7272), and Secondary antibodies, mouse and rabbit, were purchased from Santa Cruz Biotechnology (Dallas, TX, USA).

### 4.2. Cytotoxicity Assay

We used 3-(4,5-dimethylthiazol-2-yl)-2,5-diphenyltetrazolium bromide (MTT) assay to determine the cytotoxicity and cell viability of cycloastragenol (CY). In detail, HCT 116^p53+/+^ cells (1 ×  10^4^ cells/well) were seeded in a 96-well culture plate, and the concentration of cycloastragenol was diluted for 24 h, respectively. MTT (1 mg/mL) was then added for 1 h for each test concentration. Thereafter, dimethyl sulfoxide (DMSO) was added and cultured for 1 min, and optical density (OD) for cytotoxicity measurement was measured at a wavelength of 540 nm using a molecular device (Molecular Devices Co., Hercules, CA, USA) [42,43].

### 4.3. Cell Cultures

We purchased the human colorectal cancer cell line HCT116^p53+/+^ and HT-29 cells from the Center for Resistance Cell Research at Seoul National University (Seoul, Korea). In addition, HCT116^p53−/−^ cells were purchased from Professor Wonchae Choi (Kyunghee University, Korea). Cell culture methods were performed as described in previous papers [5,44].

### 4.4. p53 Stability Assay Using Cycloheximide

We first seeded HCT116^p53+/+^ cells in 6-well plates and treated with CY for 24 h. Cells were treated with 50 μM cycloheximide for 0, 15, 30, and 60 min to confirm p53 protein safety and then harvested for Western blots of p53, β-actin, and/or p21.

### 4.5. Isolation of the Total RNA and Real-Time PCR

We analyzed the transcription level of p53 in HCT116^p53+/+^ cells using real-time PCR. For a detailed method, HCT116^p53+/+^ cells were seeded on a 6-well plate (3 × 105 cells/well) and incubated for 24 h, then the cells were treated with CY (0, 25, 50 μM) according to the manufacturer’s protocol (Gen All, Seoul, Korea), and RNA was extracted using RiboEx and GenAl Hybrid-RNA purification kits. After that, RNA was quantified using NanoDrop (Thermo Fisher Scientific, Waltham, MA, USA) and used for cDNA amplification. Amplification of cDNA was performed using Maxim RT premix (inNtRON Biotech, Seongnam-si, Korea). Real-time PCR was performed using Applied Biosystems (Applied Biosystems, Foster City, CA, USA, Application Biosystems) according to [45,46,47]. The following PCR primers were used: p53, 5′-GTGACACGCTTCCCTGGATT-3′ (forward) and 5′-TTCCTGACTCAGAGGGG-3′ (reverse); p21, 5′-AAACGGGAACCAGGA-3′ (forward) and 5′-AGCGGAACAAGGAGT-3′ (reverse); PU-MA, 5′-GACGACCTCAACGCACAGTA-3′ (forward) and 5′-AGGATCCCATGATGAGATT-3′ (reverse); GAPDH, 5′-GCCACATCGTCAGACACC-3′ (forward) and 5′-CCCAATAC-GACCAAATCCGT-3′ (reverse). Primers were ordered from Bioneer (Daejeon, Korea).

### 4.6. Western Blotting

We seeded HCT116 cells and treated with CY at 0, 25, 50 and 100 μM for 24 h. Cells were then taken for each treatment concentration and protein levels were quantified using Bradford reagents (Bio-Rad, Hercules, CA, USA). After that, the protein was transferred to the nitrocellulose membrane via sodium dodecyl-polyacrylamide gel electrophoresis (SDS-PAGE), and the membrane was incubated with 1x TBST containing 0.1% Tween-20 and blocked with skimmed milk at room temperature for 1 h. The blocked membrane was incubated diluted in PARP, p53, p21, caspase 3, Puma, Bax, Cleaved Caspase 3, RPL5, p-JNK, JNK, p-ERK, ERK, p-p38, p38 or γH2AX (1:1000) primary antibodies, or β-actin (1:5000) at 4 °C, then incubated overnight. After that, the secondary antibodies, IgG-HRP, and IgG-HRP, were cultured at room temperature for 1 h, and Image Quant LAS 500 (GE Healthcare Life Sciences, Sydney, NSW, Australia) was used for measurement [48].

### 4.7. Terminal Deoxynucleotidyl Transferase-Mediated dUTP Nick end Labeling (TUNEL) Assay

We performed a TUNEL test using the protocol of the DeadEnd™ Fluorometric TUNEL System (Promega, Madison, WI, USA) to confirm apoptosis. Specifically, HCT116 cells exposed to cycloastragenol were first fixed in 4% methanol-free formaldehyde. The fixed cells were permeabilized and pre-equilibrated prior to addition of the peroxidase-conjugated detec-tion antibody and incubated with TUNEL reaction mixture (terminal deoxynucleotidyl transferase and nucleotide mixture). Afterwards, DAPI was used to stain the nuclei.

### 4.8. siRNA Transfection

We transfected HCT116 cells with control siRNA (Bioneer, Daejeon, Korea) or RPL5 siRNA (Bioneer, Daejeon, Korea) using INTERFERin™. Transfection reagent (Polyplus-transfection Inc., New York, NY, USA). The process for generating the human L5 expression construct pcDNA3-2FLAG-L5 for RPL5 is described in that paper [4]. Briefly, we waited for 15 min for a mixture of control siRNA or RPL5 siRNA (40 nM) and then treated each well. Wells were then incubated at 37 °C for 48 h for the next experiment. Thereafter, CY was treated for 24 h before harvesting.

## 5. Conclusions

In summary, our study revealed further insights into p53 activation and function in response to CY. Our data suggest that CY induced apoptosis by p53 activation in cancer cells. Furthermore, when used to treat colorectal cancer, CY has a combinational effect with either 5-FU or doxorubicin. These results suggest that CY could be useful as an anticancer drug for the treatment of colon cancer in the future.

## Figures and Tables

**Figure 1 ijms-23-15213-f001:**
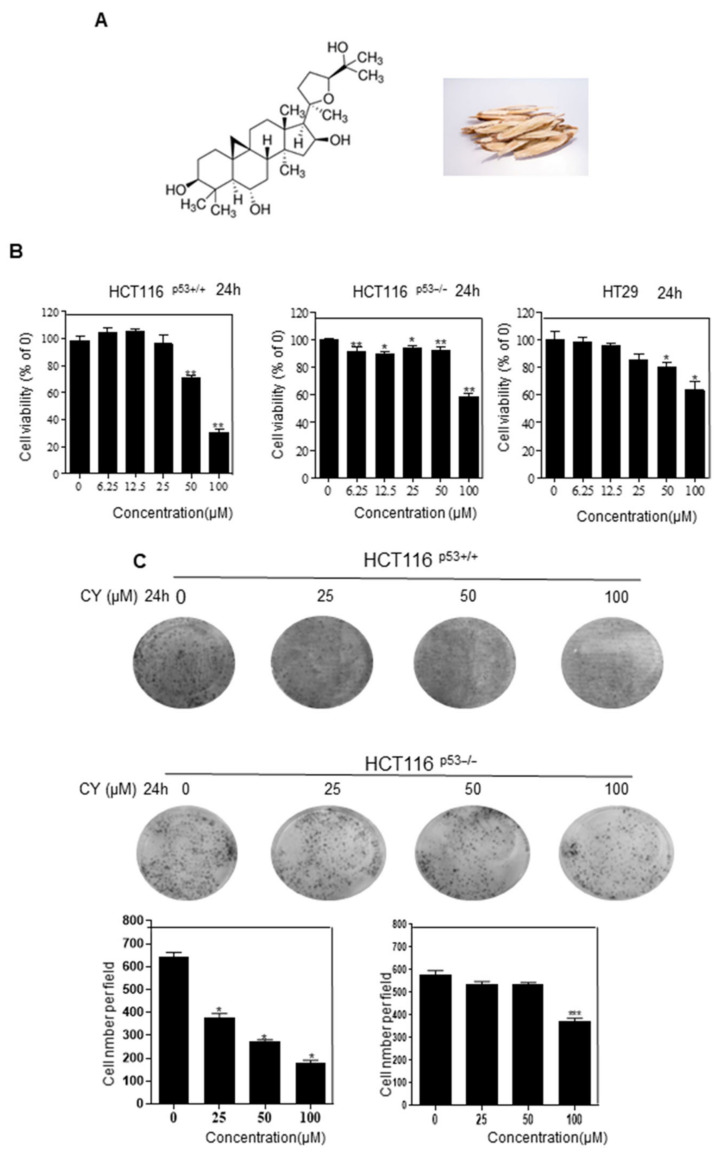
Effects of cycloastragenol (CY) on HCT116^p53+/+^, HCT116^p53−/−^ and HT29 viability. (**A**) Chemical formula of cycloastrgenol and Astragalus membranaceus Bge. the picture was shown. (**B**) Cell viability was confirmed characterized by MTT. Relative cell viability is displayed in a bar graph with comparison to the control group (100%). MTT data are expressed as mean ± S.D. * *p* < 0.05 and ** *p* < 0.01 compared to the control group. (**C**) HCT116^p53+/+^ and HCT116^p53−/−^ cells viability were confirmed with colony information. Colony information data are expressed as mean ± S.D. * *p* < 0.05 and *** *p* < 0.001 compared to the control group.

**Figure 2 ijms-23-15213-f002:**
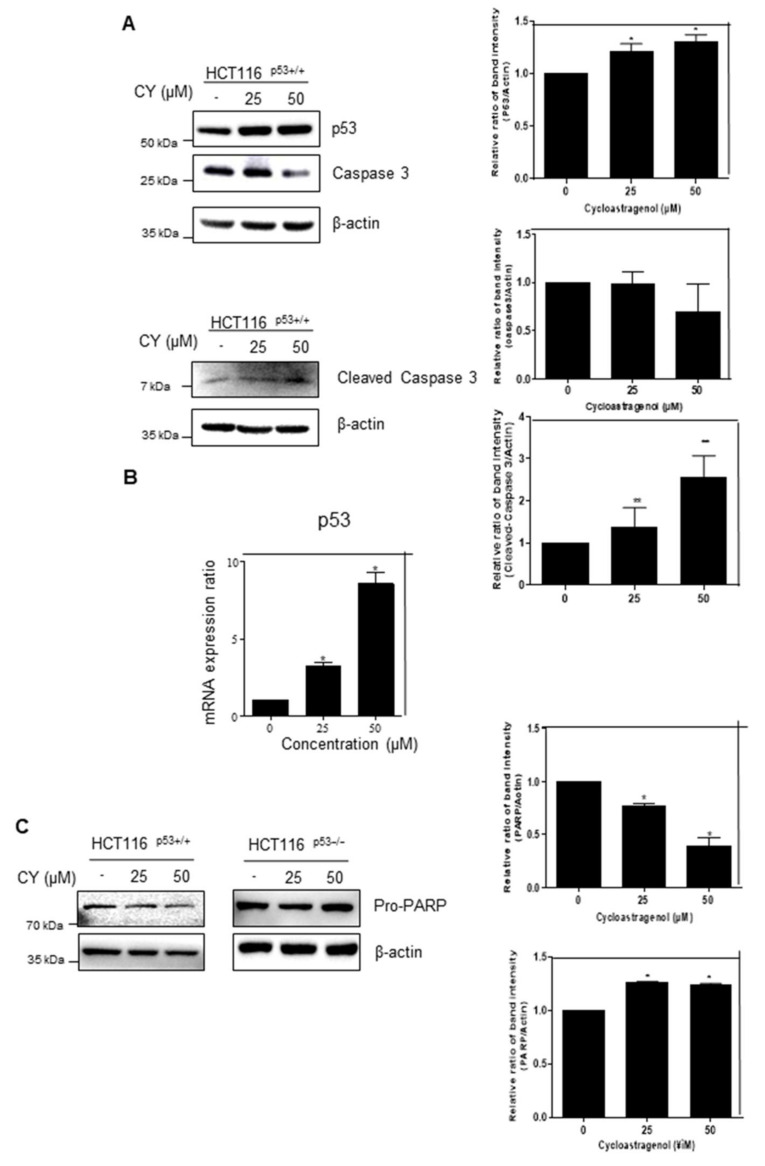
Inhibitory effect of CY on the p53 pathway in HCT116^p53+/+^ or HCT116^p53−/−^ cells. HCT116^p53+/+^ or HCT116^p53−/−^ were treated with 0, 25, and 50 μM CY for 24 h. Cells were harvested from 6 wells and extracted using cell lysis buffer. (**A**,**C**) Protein levels of p53, caspase 3, Cleaved- caspase 3, Pro-PARP and β-actin were detected by Western blotting in HCT116^p53+/+^ or HCT116^p53−/−^ cells. (**B**) The relative mRNA expression level of p53 was confirmed by reverse transcription quantitative polymerase chain reaction (qRT-PCR). mRNA expression levels were normalized to glyceraldehyde 3-phosphate dehydrogenase (GAPDH). (**D**) TUNEL assay. Relative apoptosis is displayed in a bar graph with comparison to the control group. Ratio of each protein was determined by Image J. * *p* < 0.05. ** *p* < 0.01.

**Figure 3 ijms-23-15213-f003:**
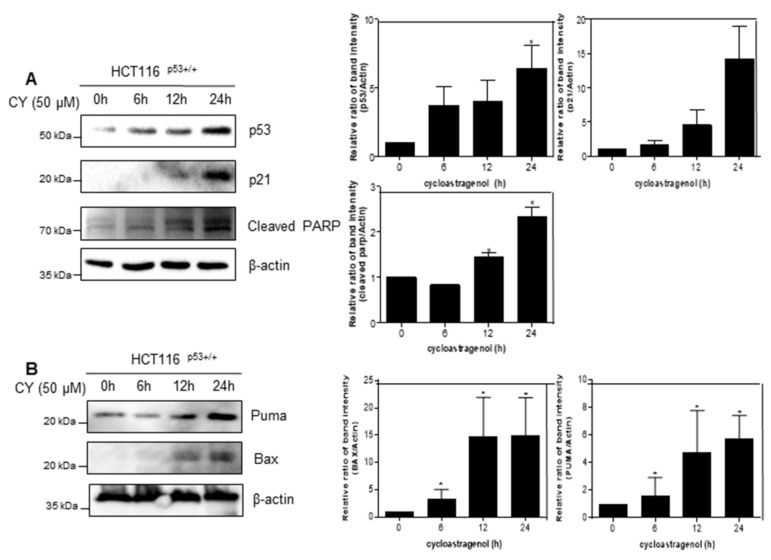
Effect of CY on expression of p53 and their target genes in HCT116^p53+/+^ cells. (**A**,**B**) HCT116^p53+/+^ cells were treated at 50 μM CY at various points (0, 6, 12, 24 h) where Western blotting was performed for p53, p21, truncated PARP, Puma, Bax, and β-actin antibodies. Ratio of each protein was determined by Image J. * *p* < 0.05.

**Figure 4 ijms-23-15213-f004:**
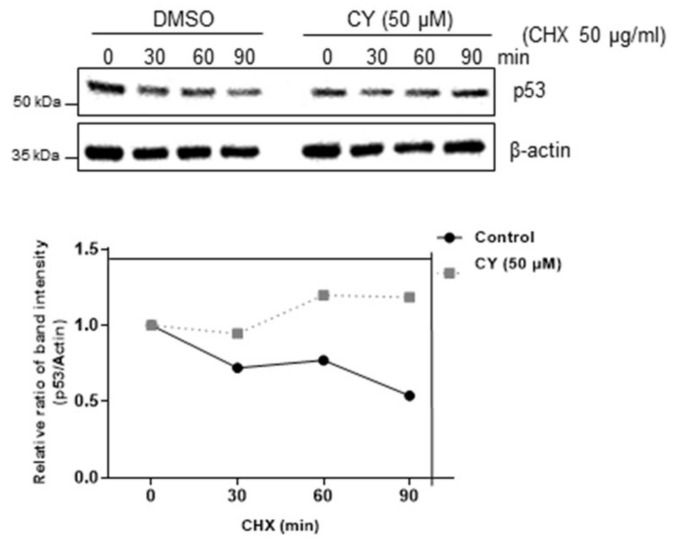
CY inhibits p53 degradation. p53 half-life was maintained in HCT116^p53+/+^ cells. HCT116^p53+/+^ cells were collected after CY treatment for 24 h for Western blotting with the indicated antibodies.

**Figure 5 ijms-23-15213-f005:**
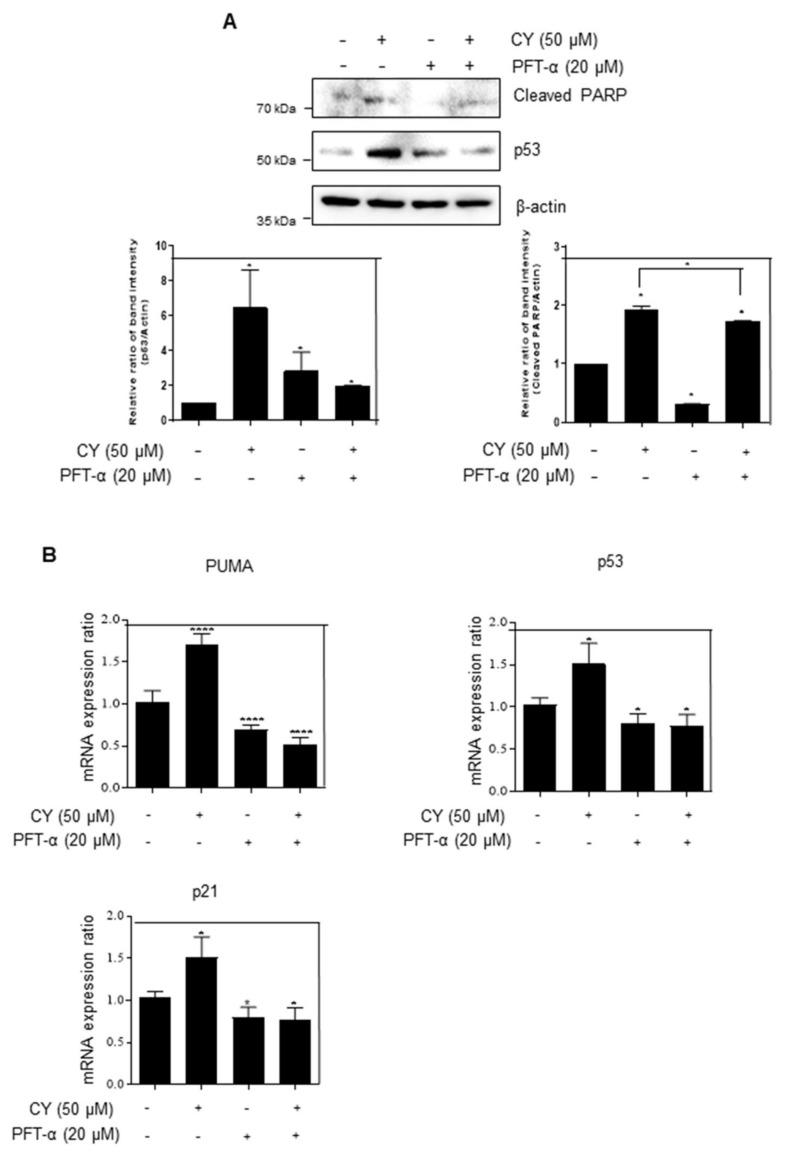
HCT116^p53+/+^ cells were treated with the p53 inhibitor PFT-α. (**A**) HCT116^p53+/+^ cells were treated with 20 μM of PFT-α for 1 h before CY 50 μM for Western blotting with p53, cleaved-PARP, and β-actin antibodies. (**B**) The relative mRNA expression level of p53, p21, PUMA were confirmed by qRT-PCR. mRNA expression levels were normalized to GAPDH. Ratio of each protein was determined by Image J. * *p* < 0.05. **** *p* < 0.0001.

**Figure 6 ijms-23-15213-f006:**
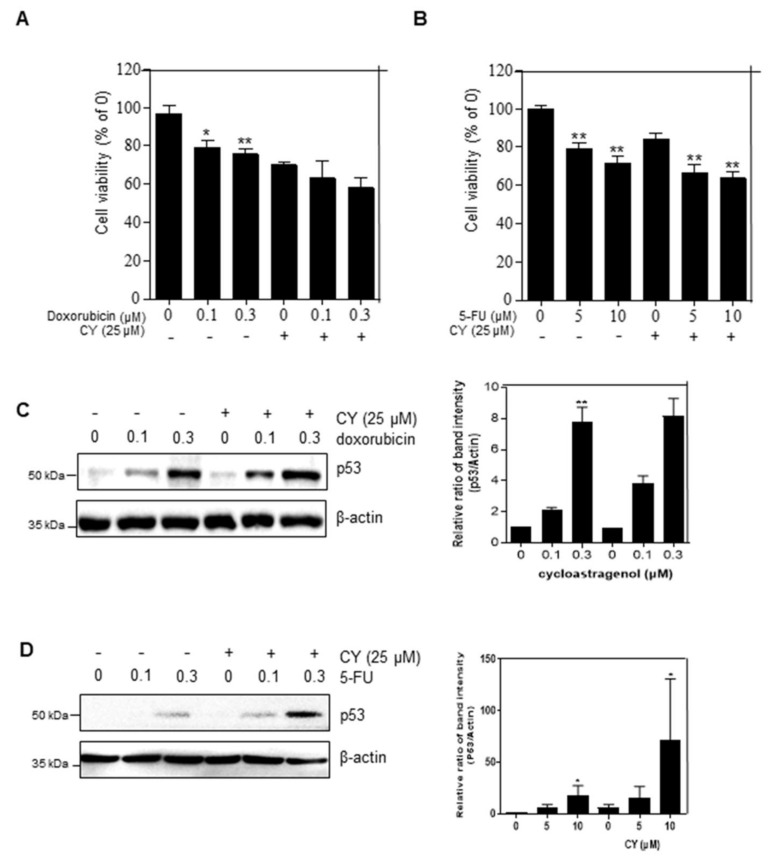
CY increases p53 expression in response to DNA-damaging drugs. (**A**,**B**) Cells were treated with CY (25 μM) and 5-FU or doxorubicin for 24 h. Cell viability was confirmed by MTT assay. Relative cell viability is displayed in a bar graph compared to the control group. The MTT data are expressed as mean ± S.D. * *p* < 0.05 and ** *p* < 0.01 compared to the control group. (**C**,**D**) HCT116^p53+/+^ cells were treated with CY (25 μM) and 5-FU (μM) or doxorubicin (μM) for 24 h. Cells were harvested and analyzed by Western blotting.

**Figure 7 ijms-23-15213-f007:**
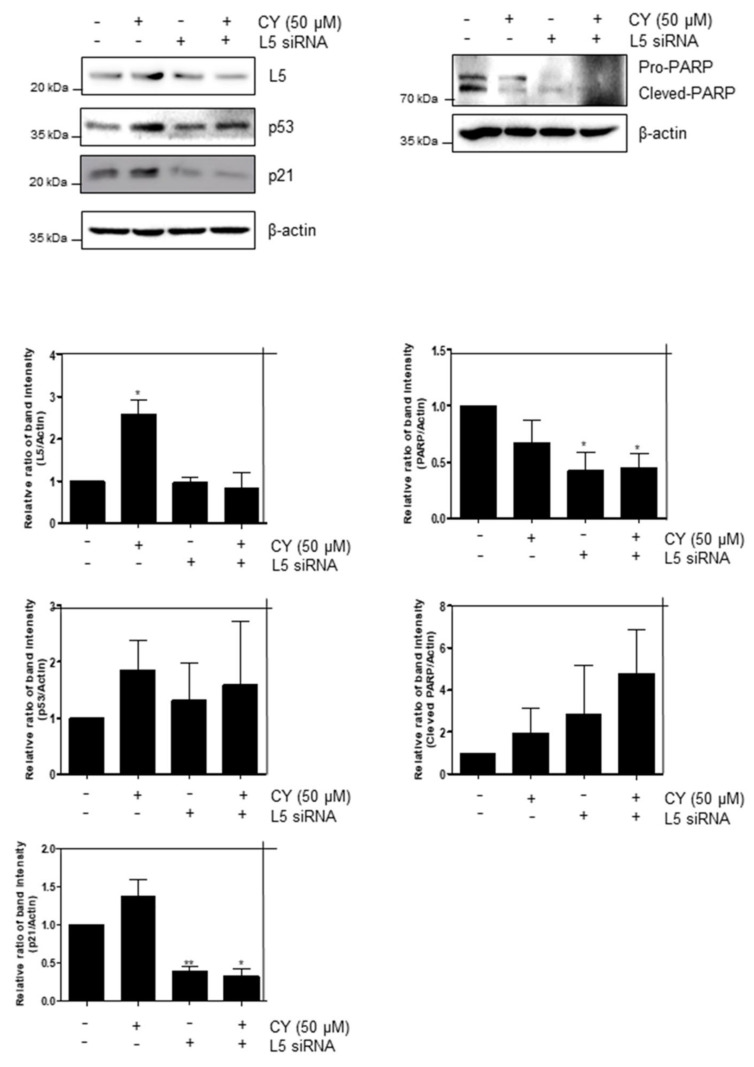
RPL5 mediates p53-dependent apoptosis in CY-treated HCT116^p53+/+^ cells. Effect of RPL5 depletion on p53, p21 and RPL5 in HCT116^p53+/+^ cells. HCT116^p53+/+^ cells were transfected with control or L5 siRNA plasmids for 48 h, exposed to CY (50 μM) for 24 h, followed by Western blotting with antibodies of p53, p21, Pro-PARP, RPL5 and β-actin carried out. Ratio of each protein was determined by ImageJ. * *p* < 0.05. ** *p* < 0.01.

**Figure 8 ijms-23-15213-f008:**
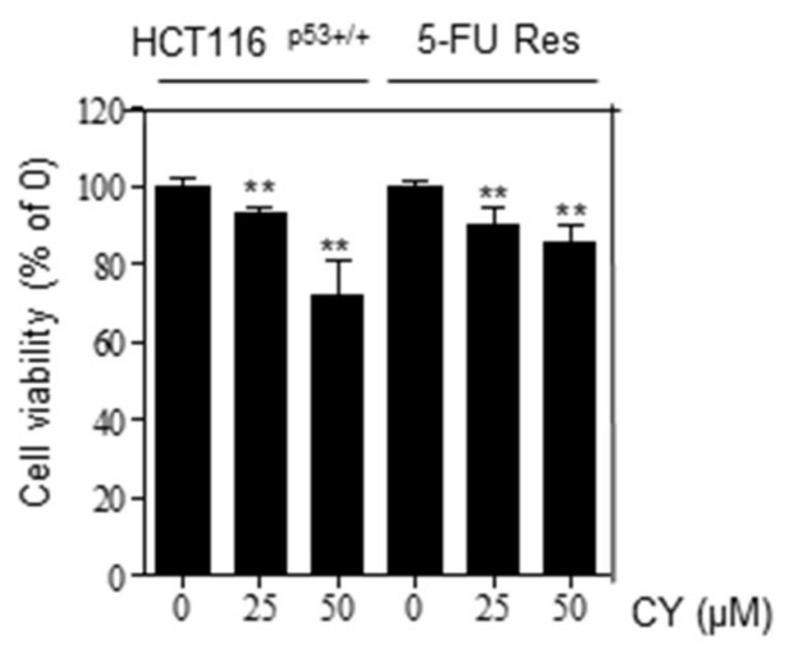
Effects of cell viability in CY-treated HCT116^p53+/+^ cells and 5-FU Res cells. Cell viability was confirmed characterized by MTT. Relative cell viability is displayed in a bar graph with comparison to the control group (100%). MTT data are expressed as mean ± S.D. ** *p* < 0.01 compared to the control group.

**Figure 9 ijms-23-15213-f009:**
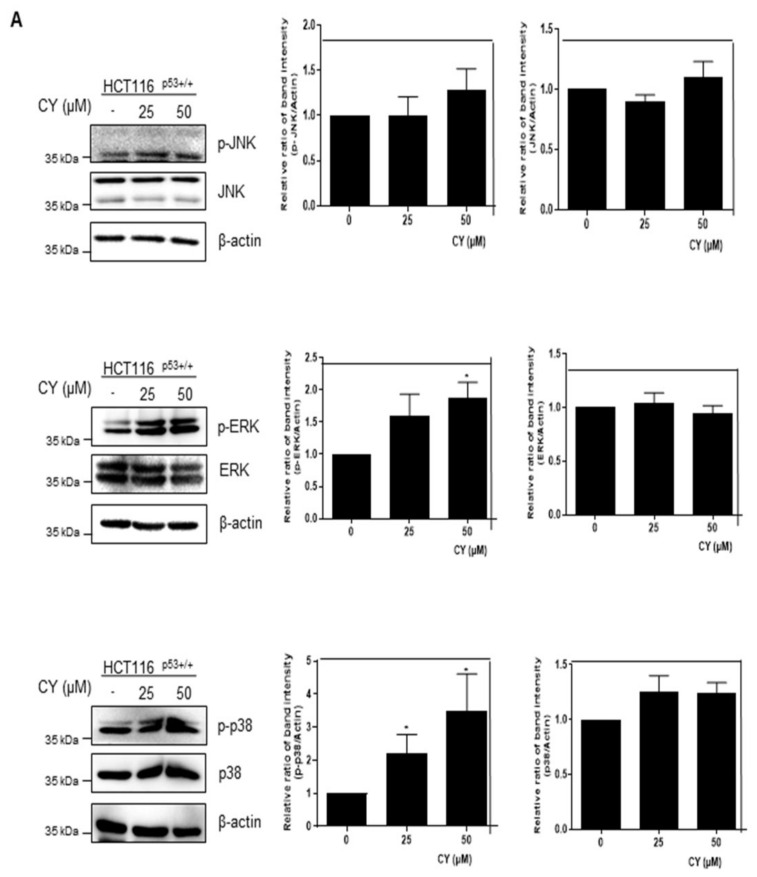
Effect of CY on expression of MAPKs and DNA damage in HCT116^p53+/+^ cells. (**A**,**C**) HCT116^p53+/+^ cells were treated with 50 μM CY at concentration 0, 25, 50 μM subjected to Western blotting for p-JNK, JNK, p-ERK, ERK, p-p38, p38, H2AX and β-actin antibodies. (**B**) HCT116^p53+/+^ cells were treated with 10 μM of SB203580 for 1 h before CY 50 μM for Western blotting with p53, p-p38, p38 and β-actin antibodies. Ratio of each protein was determined by Image J. * *p* < 0.05. ** *p* < 0.01.

**Figure 10 ijms-23-15213-f010:**
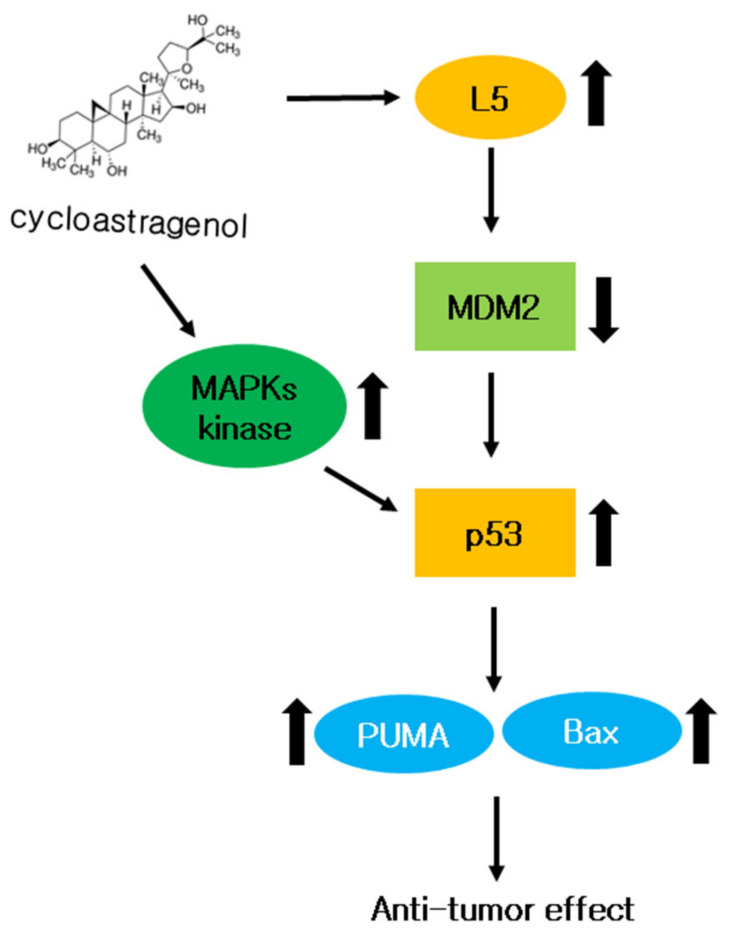
The role of CY in colon cancer cells. A mechanistic scheme showing how CY induces apoptosis by activating p53.

## Data Availability

Not applicable.

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
