# Peer review of "Antitumor Effect of Cycloastragenol in Colon Cancer Cells via p53 Activation"

_ijms, 2022, doi:10.3390/ijms232315213_

Round 1
Reviewer 1 Report
The authors showed that CY induces p53-dependent growth inhibitor and apoptosis by using p53 null HCT116 cells and their parental cells. CY accumulates p53 and induce its downstream proteins such as p21, BAX and PUMA, which promote cell cycle arrest and apoptosis (indicated by PARP cleavage and TUNEL staining) through mitochondrial pathway.
It seems to be clear that CY induce p53 accumulation and cell death; however, there is big concern in terms of the mechanism underlying p53 accumulation by CY. As authors may know, p53 is degraded by Mdm2 under physiological condition but post transcriptional modification of p53 (e.g., phosphorylation of N-terminal serine) facilitates p53 accumulation in response to stresses such as DNA damage. In this manuscript, the authors well studies the response of p53 downstream protein in cells treated with CY. On the other hand, no study was conducted for p53 upstream signaling activated by CY. If CY act as DNA damage inducer, ATM or ATR might phosphorylate p53 at Ser15 and thus accumulates p53. If CY produce intracellular reactive oxygen species, MAPK signaling might contribute to p53 signaling. Further mechanism should be uncovered before authentic publication.
Major issues
1. In Fig.1C, the result of colony formation assay in p53 KO cells was missing.
2. In Fig.2A, cleavage form of caspase-3 was missing.
3. In Fig.2D, quantification of TUNEL positive cells (e.g., FACS) is required.
4. In Fig.5, the authors used PFT-alpha to inhibit transcriptional activity of p53. But the authors did not confirmed mRNA levels of p53 downstream genes (e.g., CDKN1A coding p21 and BBC3 coding PUMA), thereby not discriminating effect of PFT-alpha on between off-target and cell death dependent on transcriptional activity of p53.
Minor points
1. Title: p53 activating -> activation
2. Line 21–22, I could not catch the meaning of "CY is expected to have a combined effect"
3. Line 14–15, the authors described "no studies have~". I think there are accumulating evidence showing the role of p53 in colon cancer.
4. Line 90, colony formation assay is not shown in Fig.2.
Author Response
Hyeung Jin Jang, Ph.D. Professor Department of Biochemistry, College of Korean Medicine Kyung Hee University, 26, KungHeedae-ro, Dondaemun-gu, Seoul, 02447, Korea E-mail: [email protected] |
Nov 15th, 2022
Reviewer-1 Round 1
The authors showed that CY induces p53-dependent growth inhibitor and apoptosis by using p53 null HCT116 cells and their parental cells. CY accumulates p53 and induce its downstream proteins such as p21, BAX and PUMA, which promote cell cycle arrest and apoptosis (indicated by PARP cleavage and TUNEL staining) through mitochondrial pathway.
It seems to be clear that CY induce p53 accumulation and cell death; however, there is big concern in terms of the mechanism underlying p53 accumulation by CY. As authors may know, p53 is degraded by Mdm2 under physiological condition but post transcriptional modification of p53 (e.g., phosphorylation of N-terminal serine) facilitates p53 accumulation in response to stresses such as DNA damage. In this manuscript, the authors well studies the response of p53 downstream protein in cells treated with CY. On the other hand, no study was conducted for p53 upstream signaling activated by CY.
Point 1: If CY act as DNA damage inducer, ATM or ATR might phosphorylate p53 at Ser15 and thus accumulates p53. If CY produce intracellular reactive oxygen species, MAPK signaling might contribute to p53 signaling. Further mechanism should be uncovered before authentic publication.
(Response) First of all, thank you for the careful review. As the reviewer pointed out, we conducted the p53 upstraem pathway experiment under the assumption that CY is a DNA damage inducer, and as a result, it was confirmed that CY was increased in MAPK siganlaing. Therefore, we added data to Figure 10 and added content to the discussion section. (line 218-230)
Major issues
Point 2: In Fig.1C, the result of colony formation assay in p53 KO cells was missing.
(Response) Sorry for any inconvenience. As the author pointed out, we added colony formation assay data in Figure 1C using p53 null type cells.
Point 3: In Fig.2A, cleavage form of caspase-3 was missing.
(Response) Sorry for any inconvenience. we added a cleavage form of caspase 3 in Figure 2A.
Point 4: In Fig.2D, quantification of TUNEL positive cells (e.g., FACS) is required.
(Response) As the authors pointed out, we added quantification of TUNEL positive cells in Figure 2D.
Point 5: In Fig.5, the authors used PFT-alpha to inhibit transcriptional activity of p53. But the authors did not confirmed mRNA levels of p53 downstream genes (e.g., CDKN1A coding p21 and BBC3 coding PUMA), thereby not discriminating effect of PFT-alpha on between off-target and cell death dependent on transcriptional activity of p53.
(Response) As pointed out by the authors, we tried to identified the mRNA expression of PUMA and p21 using PFT-alpha. As a result, it was confirmed that PUMA and p21 were decreased even when p53 transcriptional activity was suppressed. These results confirmed that p53 was activated and PUMA and p21 were activated as CY was processed. (line 121-128)
Minor points
- Title: p53 activating -> activation
(Response) Thanks for your comments. As the reviewer recommended, we replaced the word.
- Line 21–22, I could not catch the meaning of "CY is expected to have a combined effect"
(Response) Sorry for any inconvenience. we revised that it is expected to have an anti-cancer effect in line 21-22
- Line 14–15, the authors described "no studies have~". I think there are accumulating evidence showing the role of p53 in colon cancer.
(Response) Thanks for your detail comments. The story we wanted to talk about means that, there have been no cases related to the study of mechanisms of p53 using CY.
- Line 90, colony formation assay is not shown in Fig.2.
(Response) There is no colony formation explanation in line 90.
Reviewer 2 Report
In the manuscript “Antitumor effect of Cycloastragenol in colon cancer cells via p53 activating” the authors aimed to investigate the molecular mechanism underlying the cytotoxic effect of Cycloastragenol (CY) in colon cancer cells. Specifically, they demonstrated that CY inhibited cell proliferation of colon cancer cells and induced apoptosis via p53 activation.
CY is the biologically active triterpene aglycone of Astragaloside IV and it is commonly used for treating hypertension, cardiovascular disease, diabetic nephropathy, viral hepatitis, and various inflammatory-linked diseases.
In this light, the topic of the manuscript is interesting and up-to-date; the manuscript is written clearly and the presentation of results follows a coherent line; various methods and techniques are used.
However, I have some comments and critics on the current version of the manuscript to share with the Authors.
Results section
- Since western blots should be large enough to see the relevant features, I recommend to provide uncropped full-length blots showed in Figures 2-7. I suggest to include them in the Supplementary Materials and refer to them in the figure legends.
- Furthermore, I recommend to minimize the modification of the signals obtained by western blotting analysis. When it is strictly necessary to modify the image excessively, I recommend providing two different exposures of the signals in the supplementary figures.
- I recommend to add the time of CY treatment in Figure 1 legend.
- I suggest to add the percentage of TUNEL positive cells in Figure 2D.
- I suggest to improve the quality of images. Specifically, the bar diagrams are too small.
- I suggest to review the figure legends because they contain some mistakes (e.g. In Figure 1: “Cell viability was confirmed characterized by MTT.”; “Colony information” instead of “Colony formation”. In Figure 2D: the description of TUNEL staining is wrong.)
- The authors stated that ribosomal protein L5 mediates p53 activation due to CY in HCT 116 cells.
Since there are different ribosomal proteins involved in p53 activation, the authors should clarify why they chose to investigate only the ribosomal protein L5 expression upon CY treatment.
Discussion section
- Since the increased amount of specific ribosomal proteins is a hallmark of ribosomal stress response, the authors might suppose that CY is able to induce ribosomal stress. In this context, I have noticed that both drugs (5-FU and Doxorubicin) used in combination with CY are able to induce ribosomal stress. In this light, the authors should discuss the most recent findings; it could be beneficial for the discussion. There is a related paper that might be mentioned/discussed too: Carotenuto et al., Therapeutic Approaches Targeting Nucleolus in Cancer. Cells. 2019. doi: 10.3390/cells8091090.
- The ribosomal protein L5 belongs to a subset of ribosomal proteins able to exert several extra-ribosomal functions involved in the regulation of several cellular processes including cell cycle, DNA repair, maintenance of genome integrity, cellular proliferation, apoptosis, autophagy, cell migration and invasion. These aspects are extensively discussed in recent review articles that might be mentioned/discussed too: Pecoraro et al., Ribosome Biogenesis and Cancer: Overview on Ribosomal Proteins. Int J Mol Sci. 2021. doi: 10.3390/ijms22115496.
- The authors should improve the mechanistic scheme showed in Figure 9 by adding the results concerning the ribosomal protein L5.
Author Response
Hyeung Jin Jang, Ph.D. Professor Department of Biochemistry, College of Korean Medicine Kyung Hee University, 26, KungHeedae-ro, Dondaemun-gu, Seoul, 02447, Korea E-mail: [email protected] |
Nov 15th, 2022
Reviewer-2 Round 1
In the manuscript “Antitumor effect of Cycloastragenol in colon cancer cells via p53 activating” the authors aimed to investigate the molecular mechanism underlying the cytotoxic effect of Cycloastragenol (CY) in colon cancer cells. Specifically, they demonstrated that CY inhibited cell proliferation of colon cancer cells and induced apoptosis via p53 activation.
CY is the biologically active triterpene aglycone of Astragaloside IV and it is commonly used for treating hypertension, cardiovascular disease, diabetic nephropathy, viral hepatitis, and various inflammatory-linked diseases.
In this light, the topic of the manuscript is interesting and up-to-date; the manuscript is written clearly and the presentation of results follows a coherent line; various methods and techniques are used.
However, I have some comments and critics on the current version of the manuscript to share with the Authors.
Results section
Point 1: Since western blots should be large enough to see the relevant features, I recommend to provide uncropped full-length blots showed in Figures 2-7. I suggest to include them in the Supplementary Materials and refer to them in the figure legends.
(Response) Thanks for your comments. As reviewers pointed out, we showed the uncropped blots in supplementary.
Point 2: Furthermore, I recommend to minimize the modification of the signals obtained by western blotting analysis. When it is strictly necessary to modify the image excessively, I recommend providing two different exposures of the signals in the supplementary figures.
(Response) Thanks for your comments. And we are sorry if we caused you any inconvenience. We tried to put the most of the images without modification, however we corrected some of images because the data would not be clearly confirmed by the judges due to the distortion of light. So, as reviewer pointed out, we try to put the original images without modification as raw-data file.
Point 3: I recommend to add the time of CY treatment in Figure 1 legend.
(Response) Thanks for your comments. As the reviewer recommended, we added CY processing time.
Point 4: I suggest to add the percentage of TUNEL positive cells in Figure 2D.
(Response) Thanks for your comments. As the reviewer recommended, we added a percentage of TUNEL to Figure 2D.
Point 5: I suggest to improve the quality of images. Specifically, the bar diagrams are too small.
(Response) As the reviewer recommended, we made the graph larger overall from Figure 2 to Figure 7.
Point 6: I suggest to review the figure legends because they contain some mistakes (e.g. In Figure 1: “Cell viability was confirmed characterized by MTT.”; “Colony information” instead of “Colony formation”. In Figure 2D: the description of TUNEL staining is wrong.)
(Response) As recommended by the reviewer, Figure 1 legend was reviewed and revised again. Sorry for any inconvenience.
Point 7: The authors stated that ribosomal protein L5 mediates p53 activation due to CY in HCT 116 cells.
Since there are different ribosomal proteins involved in p53 activation, the authors should clarify why they chose to investigate only the ribosomal protein L5 expression upon CY treatment.
(Response) As the reviewer pointed out, L11 is also involved with p53 activation. The experiment confirmed that there was no significant different in the activity of p53 depending on the presence or absence of L11. The reason why we only showed the L5 result in our manuscript.
Discussion section
Point 8: Since the increased amount of specific ribosomal proteins is a hallmark of ribosomal stress response, the authors might suppose that CY is able to induce ribosomal stress. In this context, I have noticed that both drugs (5-FU and Doxorubicin) used in combination with CY are able to induce ribosomal stress. In this light, the authors should discuss the most recent findings; it could be beneficial for the discussion. There is a related paper that might be mentioned/discussed too: Carotenuto et al., Therapeutic Approaches Targeting Nucleolus in Cancer. Cells. 2019. doi: 10.3390/cells8091090.
(Response) We added the context in the discussion part (line 242-251) as reviewer’s recommended. Thanks for your detail comments.
Point 9: The ribosomal protein L5 belongs to a subset of ribosomal proteins able to exert several extra-ribosomal functions involved in the regulation of several cellular processes including cell cycle, DNA repair, maintenance of genome integrity, cellular proliferation, apoptosis, autophagy, cell migration and invasion. These aspects are extensively discussed in recent review articles that might be mentioned/discussed too: Pecoraro et al., Ribosome Biogenesis and Cancer: Overview on Ribosomal Proteins. Int J Mol Sci. 2021. doi: 10.3390/ijms22115496.
(Response) Citing a reference recommended by the reviewer, the ribosomal protein L5 belongs to a subset of ribosomal protection possible to extra-ribosomal functions expanded in the regulation of sub-cellular processes inclusion, DNA demonstration, reproduction, demonstration, evidence, evidence, evidence, evidence, evidence, evidence The relationship between L5 and P53 and the contents of apoptosis induction were added and described.
(line 252-259)
Point 10: The authors should improve the mechanistic scheme showed in Figure 9 by adding the results concerning the ribosomal protein L5.
(Response) As the reviewer recommended, we added related content to ribosomal protein L5 in Figure 10
Reviewer 3 Report
Attention to detail is needed as there are many places where the English is poor and this detracts from the science.
The work relies on the p53 status of the colorectal cancer cell lines – HCT116 p53 +/+ and HCT116 p53-/- cells are looked at and also HT29 cells. There is no comment on the fact HT29 cells overproduce p53 as there is a mutation in the p53 gene in these cells and hence an amino acid change in the p53 protein in these cells.
Figure 1 – the trend in cell viability is similar in HCT116 p53+/+ cells and HT29 cells following CY treatment but this should be linked to p53 status in HT29 cells. Further comment should be made about the high dose CY treatment of HCT116 p53-/- cells. The y axis labels do not make sense in Fig 1B (% of 0?). The figure title only refers to HCT116 cells and does not include HT29 cells.
The size of the bar charts in many of the figures is very small and as a consequence, the axes labels are too small to read.
Figure 2 – why have HT29 cells not been compared in this section of the study? The last two sentences of the figure legend do not seem to be relevant to this figure.
Please comment on whether p21, cleaved PARP, Puma and Bax are expressed in HCT116 p53-/- cells.
Figure 6 – why is there no p53 protein work shown for CY and 5-FU treatment?
Figure 7 – please make it clear whether the cells used in this experiment are HCT116 p53 +/+ cells.
Further clarity is needed on showing the HCT116 p53+/+ R cells are actually resistant to 5-FU. Rephrase ‘the making process was written by referring to other papers’. Rephrase ‘the group treated with HCT116 cells’.
Section 4.2 – comment on CY concentrations used.
Nomenclature consistency is needed for the HCT116 cells and their p53 status throughout the manuscript.
Section 4.7 – please explain ‘SNU-C4 cells exposed to hesperdin’.
Author Response
Hyeung Jin Jang, Ph.D. Professor Department of Biochemistry, College of Korean Medicine Kyung Hee University, 26, KungHeedae-ro, Dondaemun-gu, Seoul, 02447, Korea E-mail: [email protected] |
Nov 15th, 2022
Major revision of manuscript reference ID: ijms-1963886
Reviewer-3 Round 1
Attention to detail is needed as there are many places where the English is poor and this detracts from the science.
The work relies on the p53 status of the colorectal cancer cell lines – HCT116 p53 +/+ and HCT116 p53-/- cells are looked at and also HT29 cells. There is no comment on the fact HT29 cells overproduce p53 as there is a mutation in the p53 gene in these cells and hence an amino acid change in the p53 protein in these cells.
Figure 1 – the trend in cell viability is similar in HCT116 p53+/+ cells and HT29 cells following CY treatment but this should be linked to p53 status in HT29 cells. Further comment should be made about the high dose CY treatment of HCT116 p53-/- cells. The y axis labels do not make sense in Fig 1B (% of 0?). The figure title only refers to HCT116 cells and does not include HT29 cells.
(Response) As recommended by the reviewer, we added that the HCT116p53-/- portion does not decrease at high concentrations. (LINE 65-68) In addition, HCT116p53+/+, HCT116p53-/- and HT29 were added to the figure 1 legend.
The size of the bar charts in many of the figures is very small and as a consequence, the axes labels are too small to read.
(Response) As the reviewer pointed out, the overall size of Figure was increased.
Figure 2 – why have HT29 cells not been compared in this section of the study? The last two sentences of the figure legend do not seem to be relevant to this figure.
(Response) Thanks for your detail comments. We confirmed through MTT whether CY inhibits the growth of cancer cells in various colon cancer cells. The reason why HT29 cells were not used in subsequent experiment is that they were p53 mutation type cells.
Please comment on whether p21, cleaved PARP, Puma and Bax are expressed in HCT116 p53-/- cells.
(Response) p21, cleared PARP, PUMA, and BAX do not appear in HCT116 p53-/- cells. The reason is that these factors are the p53 WILD TYPE target factor, although they have already been identified in several papers. We will attach a reference for the reviewer to refer to.
Kaeser MD, Pebernard S, Iggo RD. Regulation of p53 stability and function in HCT116 colon cancer cells. J Biol Chem. 2004 Feb 27;279(9):7598-605. doi: 10.1074/jbc.M311732200. Epub 2003 Dec 9. PMID: 14665630.
Yu J, Zhang L, Hwang PM, Kinzler KW, Vogelstein B. PUMA induces the rapid apoptosis of colorectal cancer cells. Mol Cell. 2001 Mar;7(3):673-82. doi: 10.1016/s1097-2765(01)00213-1. PMID: 11463391.
Figure 6 – why is there no p53 protein work shown for CY and 5-FU treatment?
(Response) As pointed out by the reviewer, 5-FU and CY combination processing data were added to Figure 6.
Figure 7 – please make it clear whether the cells used in this experiment are HCT116 p53 +/+ cells.
(Response) As recommended by the reviewer, the Figure 7 legend part was modified to HCT116p53 +/+.
Further clarity is needed on showing the HCT116 p53+/+ R cells are actually resistant to 5-FU. Rephrase ‘the making process was written by referring to other papers’. Rephrase ‘the group treated with HCT116 cells’.
(Response) As recommended by the author, it is clearly described in Line 175-178.
Section 4.2 – comment on CY concentrations used.
Nomenclature consistency is needed for the HCT116 cells and their p53 status throughout the manuscript.
(Response) As recommended by the author, we have consistently modified the p53 status in HCT116 cells as a whole.
Section 4.7 – please explain ‘SNU-C4 cells exposed to hesperdin’.
(Response) We are not able to find the word “Hesperdin” in our paper.
Round 2
Reviewer 1 Report
The authors responded to almost all of my comments but a major issue has not yet been solved.
The authors insist that CY induces DNA damage, thereby inducing p38 MAPK activation and following p53 accumulation. In fact, both p38 MAPK phosphorylation and p53 accumulation were observed in CY-treated cells.
However, there is no direct evidence demonstrating that p38 MAPK phosphorylation induced by CY contribute to p53 accumulation. To demonstrate it, the authors should confirm inhibition of p38 signaling (e.g., by using SB203580) decrease the abundance of p53.
In addition, there is no evidence that CY induce DNA damage. The type of DNA damage is also unclear. Does CY induce DNA single strand break?, double strand break? , replicative stress?, 8-oxoguanin?
For example, strand break is detected by comet assay or pulse field gel electrophoresis. Replicative stress is detected by DNA fiber assay. 8-oxoguanin is detected by ELISA.
Author Response
Hyeung Jin Jang, Ph.D. Professor Department of Biochemistry, College of Korean Medicine Kyung Hee University, 26, KungHeedae-ro, Dondaemun-gu, Seoul, 02447, Korea E-mail: [email protected] |
Nov 28th, 2022
Reviewer-1 Round 2
The authors responded to almost all of my comments but a major issue has not yet been solved.
The authors insist that CY induces DNA damage, thereby inducing p38 MAPK activation and following p53 accumulation. In fact, both p38 MAPK phosphorylation and p53 accumulation were observed in CY-treated cells.
However, there is no direct evidence demonstrating that p38 MAPK phosphorylation induced by CY contribute to p53 accumulation. To demonstrate it, the authors should confirm inhibition of p38 signaling (e.g., by using SB203580) decrease the abundance of p53.
In addition, there is no evidence that CY induce DNA damage. The type of DNA damage is also unclear. Does CY induce DNA single strand break?, double strand break? , replicative stress?, 8-oxoguanin?
For example, strand break is detected by comet assay or pulse field gel electrophoresis. Replicative stress is detected by DNA fiber assay. 8-oxoguanin is detected by ELISA.
Results section
Point 1: The authors responded to almost all of my comments but a major issue has not yet been solved.
The authors insist that CY induces DNA damage, thereby inducing p38 MAPK activation and following p53 accumulation. In fact, both p38 MAPK phosphorylation and p53 accumulation were observed in CY-treated cells.
However, there is no direct evidence demonstrating that p38 MAPK phosphorylation induced by CY contribute to p53 accumulation. To demonstrate it, the authors should confirm inhibition of p38 signaling (e.g., by using SB203580) decrease the abundance of p53.
(Response) As noted by the reviewer, we tested whether p38 MAPK affects the direct accumulation of p53 using p38 inhibitor (SB203580). As a result of the experiment, it was confirmed that the activity of p53 is affected by p38 (Figure 9B).
Point 2: In addition, there is no evidence that CY induce DNA damage. The type of DNA damage is also unclear. Does CY induce DNA single strand break?, double strand break? , replicative stress?, 8-oxoguanin?
For example, strand break is detected by comet assay or pulse field gel electrophoresis. Replicative stress is detected by DNA fiber assay. 8-oxoguanin is detected by ELISA.
(Response) As the reviewer pointed out, we tested pH2AX protein expression (a representative DNA damage marker). The data showed that CY induces phospho-H2AX protein expression (Figure 9C) (Line 241-245).
Round 3
Reviewer 1 Report
The authors demonstrated p38 MAPK plays a pivotal role for CY-induced p53 accumulation. In addition, they also demonstrated that CY elicited DNA damage indicated by phospho-H2AX. The additional data improved manuscript and thus seems to be acceptable. Please note that H2AX phosphorylation is induced by apoptotic signaling as well as DNA damage. To discriminate these, pan-caspase 3 inhibitor such as Z-VAD FMK or DNA-PK inhibitor such as NU7026 are useful (Please see following article: DOI: 10.1128/mcb.00581-08 ). I hope that the authors clarify whether the H2AX phosphorylation is a result of CY-induced DNA damage or not in the future study.